# A High-Performance Digital Interface Circuit for a High-Q Micro-Electromechanical System Accelerometer

**DOI:** 10.3390/mi9120675

**Published:** 2018-12-19

**Authors:** Xiangyu Li, Jianping Hu, Xiaowei Liu

**Affiliations:** 1Faculty of Information Science and Technology, Ningbo University, Ningbo 315211, China; lixiangyu@nbu.edu.cn; 2MEMS Center, Harbin Institute of Technology, Harbin 150001, China; liuxiaowei3@outlook.com

**Keywords:** MEMS, interface circuit, high-Q capacitive accelerometer, Sigma-Delta

## Abstract

Micro-electromechanical system (MEMS) accelerometers are widely used in the inertial navigation and nanosatellites field. A high-performance digital interface circuit for a high-Q MEMS micro-accelerometer is presented in this work. The mechanical noise of the MEMS accelerometer is decreased by the application of a vacuum-packaged sensitive element. The quantization noise in the baseband of the interface circuit is greatly suppressed by a 4th-order loop shaping. The digital output is attained by the interface circuit based on a low-noise front-end charge-amplifier and a 4th-order Sigma-Delta (ΣΔ) modulator. The stability of high-order ΣΔ was studied by the root locus method. The gain of the integrators was reduced by using the proportional scaling technique. The low-noise front-end detection circuit was proposed with the correlated double sampling (CDS) technique to eliminate the 1/*f* noise and offset. The digital interface circuit was implemented by 0.35 μm complementary metal-oxide-semiconductor (CMOS) technology. The high-performance digital accelerometer system was implemented by double chip integration and the active interface circuit area was about 3.3 mm × 3.5 mm. The high-Q MEMS accelerometer system consumed 10 mW from a single 5 V supply at a sampling frequency of 250 kHz. The micro-accelerometer system could achieve a third harmonic distortion of −98 dB and an average noise floor in low-frequency range of less than −140 dBV; a resolution of 0.48 μg/Hz^1/2^ (@300 Hz); a bias stability of 18 μg by the Allen variance program in MATLAB.

## 1. Introduction

Capacitive accelerometers are widely used in the military and civilian fields because of their low power consumption, simple structure, good stability and easy integration with the complementary metal-oxide-semiconductor (CMOS) process [1]. In recent years, high-performance capacitive accelerometers with an accuracy of sub-μg level occupy a large market share in inertial navigation, space microgravity measurement, platform stability control and other fields. The micro-accelerometers with an open-loop output have a simple structure, but the signal bandwidth is limited by the sensitive structure and the input range of the signal is greatly reduced [2,3,4]. Therefore, the micro-accelerometers usually work in a closed-loop feedback state to obtain better linearity, dynamic range and signal bandwidth. The closed-loop working mode can also increase the electrical damping of the mechanical structure and improve effectively its electrical response [5,6]. High over sampling rate (OSR), high-order topology and multi-bit quantization are used to improve the noise shaping ability of Sigma-Delta (ΣΔ) micro-accelerometers.

A large OSR requires high sampling frequency, which leads to coupling between different noise sources and increasing power consumption in ΣΔ micro-accelerometers. The high-Q sensitive structure introduces a large phase shift at the resonance frequency, and the stability of the whole high order system will be greatly reduced. If a high-Q sensitive structure is used to reduce mechanical noise and a high-order structure is used to reduce quantization noise, the problem of system stability will become a major problem. It is necessary that cascading a phase compensator after the front-end charge amplifier to provide additional phase compensation, which is equivalent to providing electrical damping to the under-damped mechanical structure to stabilize the loop. In other literature, phase compensators are placed in the feedback loop, which can improve the feedforward path gain, but this can also reduce the gain in the feedback path and reduce the input dynamic range. It is difficult to design a linear micro-accelerometer with a multi-bit quantization structure because the signal conversion process of the sensitive structure is nonlinear. At present, the main research on the interface circuit of micro-accelerometers is still based on a low-Q sensitive structure, low-order ΣΔ system and a one-bit feedback structure [7,8]. The micro-accelerometers with analog output can achieve a high precision output of less than 1 μg/Hz^1/2^, but the performance of digital closed-loop micro-accelerometers reported is difficult to achieve a precision at the sub-μg level [9]. The digital micro-accelerometers with sub-μg precision output has a lot of application requirements in the field of geophone, national defense and military. Therefore, the noise theory, system stability analysis and key technology of high-precision closed-loop micro-accelerometers are mainly studied in this paper, which is aimed at realizing a high-performance interface circuit chip with sub-μg accuracy.

The high-Q accelerometer sensitive element, front-end charge sensing circuit, sample and hold circuit, phase compensation circuit and high-order Sigma-Delta modulator circuit are introduced and designed in Section 2. In Section 3, we show a detailed analysis based on the noise characteristics and stability of micro-accelerometers with an application specific integrated circuit (ASIC) interface. The performance can be improved by a correlated double sampling (CDS) technique and a proportional scaling technique. The performance parameters of micro-accelerometers were tested by the experiments. Finally, Section 4 concludes the study of a high-Q MEMS accelerometer with a high-precision integrated circuit and testing results, which show that the performance level of micro-accelerometers in this work has great advantages in the application of inertial navigation and the nano-satellites field by comparison.

## 2. Materials and Methods

### 2.1. Materials

The high-Q sensitive structure which is encapsulated in vacuum is from Colibrys Company (Neuchatel, Switzerland). The interface circuit based on micro-accelerometers was fabricated by a 0.35 μm CMOS process and cooperated with Shanghai Huahong Integrated Circuit (Shanghai, China).

### 2.2. High-Q Accelerometer Sensitive Element

The equivalent bridge model of the vacuum packaged silicon micro-accelerometers is shown in Figure 1. The upper and lower capacitance plates in Figure 1 are fixed plates and the equivalent variable capacitors *C_S_*_1_ and *C_S_*_2_ are formed between the mass and the plates. *C_P_*_1_ and *C_P_*_2_ are parasitic capacitors. When the external acceleration acts on the sensitive element, the displacement of the mass will change, which is relative to the plates. This can result in the corresponding change of the variable capacitance. The change of the two equivalent sensitive capacitances will be perceived by the post-detection circuit. The accelerometer sensitive element with vacuum packaged silicon structure used for design, simulation and test in this paper was obtained from Colibrys Company (SF1500). The sensitive element could achieve an open-loop resonant frequency of 1 kHz, a high-quality factor of more than 30 and a Brownian noise corresponding of an equivalent acceleration of less than 60 ng/Hz^1/2^. The corresponding static capacitance and the sensitivity of the sensor element were 180 pF and 10 pF/g. Major parameter indicators are shown as in Table 1.

Figure 2 shows the differential capacitance model of the sensitive structure, in which *d* was the distance between the upper and lower plates. When the mass is in equilibrium and the two differential capacitance values are equal, the static capacitance is shown as follows:(1)$$C_{0}=\frac{\varepsilon \varepsilon_{0}A}{d}$$

*ε*_0_—the vacuum dielectric constant

*ε*—the relative dielectric constant between the sensitive capacitor plates

*A*—the positive area of the sensitive capacitor plates

*x*—the displacement of the sensitive mass block under the external acceleration

When the displacement of the sensitive mass causes changes in differential capacitance pairs, the variable capacitance *C_S_*_1_ and *C_S_*_2_ in Figure 2 can be expressed respectively:(2)$$C_{S1}=\frac{\varepsilon \varepsilon_{0}A}{d-x}=\frac{C_{0}}{1-\frac{x}{d}}$$
(3)$$C_{S2}=\frac{\varepsilon \varepsilon_{0}A}{d+x}=\frac{C_{0}}{1+\frac{x}{d}}$$

In the closed-loop system, the displacement of the sensitive mass was very small relative to the plate spacing. The relative variation of the capacitance (Δ*C*) can be written as follows:(4)$$\Delta C=C_{S1}-C_{S2}=\frac{\varepsilon \varepsilon_{0}A}{d-x}-\frac{\varepsilon \varepsilon_{0}A}{d+x}\approx 2C_{0}\frac{x}{d}$$

It can be seen that the relative displacement of the mass and the input acceleration signal are approximately linear in the input signal band, which was much smaller than the resonant frequency of the mechanical structure. That is $x\approx \frac{a}{\omega_{0}^{2}}$, where *a* denotes the acceleration signal and ω_0_ denotes the mechanical resonance frequency. The Equation (4) can be expressed as:(5)$$a=\frac{\Delta Cd\omega_{0}{}^{2}}{2C_{0}}$$

### 2.3. High-Order Interface Circuit Based on Micro-Accelerometers

The closed-loop micro-accelerometers use the feedback principle of electrostatic force to confine the sensitive mass to the balance position, which greatly reduces the sensitive mass’s displacement in order to reduce the nonlinear error in charge conversion and improve the overall linearity, bandwidth and amplitude range of the input acceleration signal. In this paper we propose a high-precision digital micro-accelerometer with sub-μg noise level with a high-Q sensitive structure encapsulated in a vacuum, which was used to reduce the mechanical noise. The high-order noise shaping ability was realized by combining the high-order topological structure. Due to underdamping, slow corresponding output in response and poor seismic performance of high vacuum mechanical structures, there will be stability problems after constituting a high-order system with the ΣΔ modulator. Therefore, when the system of micro-accelerometers can achieve sub-μg noise level, the stability of the system should be fully considered in the digital interface circuit design of micro-accelerometers [10,11,12,13]. Aiming at the stability problem of high-precision digital micro-accelerometer interface circuit, a phase compensator circuit can be designed to provide phase compensation, enhance electrical damping and improve the system response. In addition, in order to overcome the influence of process error on the stability of high-order interface circuit, reasonable circuit design and parameter optimization are needed.

Figure 3 shows a diagram of the front-stage charge-sensitive circuit. In this paper, we propose a fully differential switched-capacitor detection circuit, in which *C_R_* is the reference capacitor and *C_f_* is the integral capacitor. The front-stage sensing circuit consists of an equivalent mechanical structure, a reference capacitor pair, a charge-sensitive and a correlated double-sampling and holding module. The output voltage of the charge sensitive circuit can be expressed as:(6)$$V_{out}=\frac{2V_{r}\Delta C}{C_{f}}=\frac{4C_{0}V_{r}}{C_{f}d\omega_{0}^{2}}a(f)$$

The input acceleration signal is converted into the voltage signal of the front-stage sensitive circuit. In Equation (6), *V_r_* is the reference voltage. The sensitivity of the detection was limited by the initial capacitance of the sensitive structure, the distance between the plates and the resonant frequency of the mechanical structure. In this paper the static capacitance value of the sensitive structure and the reference capacitance were 180 pF respectively. An additional capacitor can be connected in parallel with the sensitive structure to increase the equivalent static capacitance value. But the static capacitance can’t be increased indefinitely, which will affect the loop stability and the accuracy of charge conversion. The timing diagram of the front-end circuit is as shown in Figure 3b. There are five phases in operation of the circuit, which is the reset phase, charge sensing phase A, charge sensing phase B, sampling phase and electrostatic force feedback phase. The switch S4_inv and S5_inv were the reverse clock of S4 and S5, respectively. Electrostatic force feedback and charge sensitivity operate at different times of a cycle to eliminate noise coupling between them. In the reset phase, the input electrode voltage of the interface was reset to ensure a correct bias point and the capacitor was discharged to erase the memory from the previous cycle. A small size of switch S6 was designed to reduce charge injection. In the charge sensing phase A, the reference voltages +*V_s_* and −*V_s_* were applied to the sensor mass and common electrode of the reference capacitors, respectively. The capacitor stores the amplified voltage and the error signal including the offset and noise of the operational amplifier. The output of the charge sensing is given by:(7)$$\Delta V_{out1}=V_{error}-V_{S}\frac{C_{S1}-C_{S2}}{C_{f}}$$
where *C_f_* is the integration capacitance (10 pF). During the charge sensing phase B, the voltages of sensor mass and common electrode of the reference capacitors were kept at +*V_s_* and −*V_s_*, respectively. The output of the charge sensing is expressed as:(8)$$\Delta V_{out2}=V_{error}+V_{S}\frac{C_{S1}-C_{S2}}{C_{f}}$$

The differential output of the sample and hold circuit is represented by:(9)$$\Delta V_{out}=\Delta V_{out2}-\Delta V_{out1}=2V_{S}\frac{C_{S1}-C_{S2}}{C_{f}}$$

The values of the nominal capacitance of the sensor element and the reference capacitance were 180 pF. The integration capacitance was set to 10 pF, which was a trade-off between the noise performance and system stability. We set a pre-stage gain of 30 V/g and an accelerometer system sensitivity of 1.866 V/g. In this paper the bandwidth of the accelerometer was 300 Hz, which was defined by an increasing low-frequency noise spectral density of 3 dB.

The high-Q sensitive structure can introduce a pair of complex poles near the imaginary axis to the closed-loop filter. The high-frequency parasitic resonant modes and the complex poles can destabilize the high-Q system easily. In this paper we propose a phase compensator circuit which can introduce an extra zero to compensate for loop filters. The low-frequency loop gain control was considered based on a good noise shaping ability. In this lead compensator circuit, *C*_1_ and *C*_3_ had the same capacitance value. The ratio between *C*_2_ and *C*_3_ determined the compensation degree. For a high-Q sensitive structure, a heavy compensation was chosen. The sampling frequency of the phase compensator circuit was 250 kHz. The lead compensator with a transfer function in discrete-time z-domain can be expressed as:(10)$$H_{cmp}(z)=\frac{C_{1}}{C_{3}}-\frac{C_{2}}{C_{3}}z^{-1}$$

*C*_1_ and *C*_3_ have the same capacitance value and at the case of *C*_2_ = α*C*_3_, the Equation (10) can be expressed as:(11)$$H_{cmp}(z)=1-\alpha z^{-1}$$

In Equation (11), α indicates the depth of compensation. The lead compensator operates as a proportion-derivative (PD) controller and the stability is improved by positioning the zero closer to the open-loop poles of the filter, which is resulting in an increase of the amount of phase lead. If the compensation depth is insufficient or excessive, the closed-loop system may have stability problems. For over-compensated sigma-delta accelerometer systems, the system may also be unstable if the loop gain is too small. Overcompensation of sigma-delta accelerometer systems can also affect the noise shaping ability of a post-stage modulator. Although the noise shaping ability of the modulator decreases with the increase of compensation depth, more-order structure and a high-Q sensitive structure can be used to reduce the impact of the reduction of noise shaping ability caused by depth compensation. Because of the high-order system structure in this paper, we proposed a lead compensator circuit as shown in Figure 4. The stability of the system was more important than the noise shaping ability of the modulator, so we set a depth compensation coefficient of 0.9.

We propose the system structure of the ΣΔ modulator as shown in Figure 5a based on stability analysis of ΣΔ micro-accelerometers. In order to achieve a better noise suppression performance at low-frequency, we used a correlated double sampling technique to improve the noise level of the first stage integrator. The one-bit quantizer was achieved by the dynamic comparator. The output of the comparator was as a control signal to control feedback reference voltage *V_ref+_* and *V_ref−_* in the first stage integrator [14,15]. As shown in Figure 5b, the timing diagram of the ΣΔ modulator circuit, wherein *ck*1 and *ck*2 were the two-phase non-overlapping clock, *ck*1 was active-high, *ck*2 was active-low. The shutdown time of *ck*1*d* was later than *ck*1; the shutdown time of *ck*2*d* was later than *ck*2. This could effectively suppress the influence of charge injection and clock-feedthrough in the switched-capacitor (SC) circuit. In the ΣΔ modulator circuit, the double sampling technique was also used to increase the equivalent sampling frequency in order that the sampling capacitance of the input signal and the sampling capacitance of the feedback signal were separated. The charge transfer at the integration phase is reduced and the accuracy of the integrator can be improved. In this paper we propose a topology of distributed feedback ΣΔ accelerometers with a feedforward structure. This structure combines some advantages of a feedforward and feedback topology structure and has the characteristics of good system stability and a small output signal swing. We designed the main parameters of the ΣΔ modulator as shown in Table 2.

## 3. Result and Discussion

### 3.1. Noise Characteristics and Stability Analysis of Micro-Accelerometers

In consideration of a relatively low gain at low-frequency in the feedback structure and a relatively large nonlinearity problem of the output signal. Figure 6 shows the analysis model of ΣΔ micro-accelerometers in this paper. *K_x/V_* in Figure 6 is the amplification factor from the displacement output of the sensitive structure to the output voltage of the charge sensitive circuit. *H_c_* is the pre-stage phase compensator; *f*_a1_, *f_a_*_2_, *f_a_*_3_ and *f_a_*_4_ are feedforward coefficients; *f_b_*_1_, *f_b_*_2_, *f_b_*_3_ and *f_b_*_4_ are feedback coefficients; *k*_1_, *k*_2_, *k*_3_ and *k*_4_ are integrator gain coefficients; *K_V/a_* is the gain coefficient from feedback voltage to equivalent acceleration. The main noise sources introduced in the model are the Brownian noise of mechanical structure, the electrical noise of the pre-stage charge amplifier and the quantization noise of the post-stage ΣΔ. In consideration of the accuracy discreteness of the micro-accelerometer sensitive structure, there are four distributed feedback factors in the post-stage modulator circuit of the ΣΔ micro-accelerometer system in this paper. The stability of the loop can be effectively controlled by adjusting the feedback coefficient, especially adjusting the feedback coefficient *f_b_*_1_ of the first integrator. So, the local feedback factor *f_b_*_1_ is designed as an off-chip adjustable part. The low-frequency loop gain can be easily controlled to eliminate the impact of process errors and the high-order interface circuit can be applied to a different mechanical structure.

Based on the analytical model of ΣΔ micro-accelerometers, we derived the signal transfer function (STF) and noise transfer function (NTF) of the ΣΔ accelerometer system. The output swing of the integrators decreased when the gain of the integrators was reduced by using the proportional scaling technique. In this way, the reduction of the swing amplitude associated with the nonlinearity of the amplifier gain will lead to the reduction of the output harmonic distortion and the overall power consumption. The loop stability is ensured by controlling the zero-pole distribution of the loop filter to make sure that the average frequency response amplitude of noise transfer function is within a reasonable range. The values of feedforward coefficients, feedback coefficients and integrator gain coefficients were determined as shown in Table 3.

In order to stabilize the system in the high-order structure, a pre-compensator as shown in Figure 4 was added to the loop to delay the phase intersection to the gain intersection. Because the gain intersection point was very far in the high-order structure, the pre-compensator needed to provide a larger pre-phase, which required a larger compensation depth α. The increase of α will decrease the low-frequency gain, but will not affect the noise characteristics of higher-order structures. In the high-Q ΣΔ micro-accelerometers, the stability of higher order systems is strongly affected by compensation depth α. Only when α is greater than a certain critical value, the system can reach a stable state. Additionally, with the increase of Q-value, the higher order system stability requires a larger value of α. In this paper, the stability of the Sigma-Delta modulator was studied by the root locus method. The pole position of transfer function was changed by the gain of quantizer. The gain of quantizer was changed by the amplitude of the input signal. The root locus of the topology analysis model of the Sigma-Delta modulator designed is as shown in Figure 7. As the input signal amplitude increased, the quantizer gain decreased. It can be seen that from Figure 7 when the quantizer gain is more than 0.547, the root locus begins to deviate from the unit circle, which can lead to an increase in the amplitude of the input signal of the quantizer.

System parameters are optimized by improving stability and reducing harmonic distortion. The reference voltage of simulation was (±2.5 V). When the sampling frequency was 250 kHz, there was an equivalent acceleration signal amplitude of 1 g and a frequency of 30.5175 Hz. Figure 8 shows the output transient waveforms of the first-stage integrator, the second-stage integrator, the third-stage integrator and the fourth-stage integrator in sequence from top to bottom. It can be seen from Figure 8 that the output amplitude of the integrators was within a very small range of ±0.2 V. It shows that the topology of the Sigma-Delta modulator designed in this paper has the advantage of small output swing and good stability.

### 3.2. The Test of Digital Micro-Accelerometers

The ΣΔ modulator interface circuit for micro-accelerometers was fabricated in a standard 0.35 μm four layers metal double polycrystal CMOS process and the printed circuit board (PCB) photograph of the digital micro-accelerometer system is shown in Figure 9. The photograph of the interface circuit chip is also shown in Figure 9, which has 28 pins for the chip test. The active area of the chip was 3.3 mm × 3.5 mm. The 5 V power supply of the interface circuit combined with the sensitive element was supported by the Agilent E3631 (Agilent Technologies Inc, Santa Clara, CA, USA). The input signal (240 Hz) and clock signal was supplied by the Tektronix AFG3102 function signal generator (Tek Technology Co., Shanghai, China). The 65536-point digital output sequence of ΣΔ micro-accelerometers was captured by an Agilent Logic analyzer 16804A (Agilent Technologies Inc, Santa Clara, CA, USA). The ouput digital signal is used to calculate the output power spectral density (PSD) as shown in Figure 9a by a MATLAB program (R2016a, MathWorks, Natick, MA, USA).

The power dissipation of the micro-accelerometer system was 10 mW at a sampling frequency of 250 kHz. The full scale range was ±1 g and the ΣΔ modulator had a dynamic range (DR) of 97 dB. The third harmonic distortion can be calculated by the difference between the signal-to-noise ratio of the fundamental wave and signal-to-noise ratio of the third harmonic wave in the spectrogram. The ΣΔ micro-accelerometer system can achieve a third harmonic distortion of −98 dB as shown in Figure 10a and a resulting signal-to-noise ratio (SNR) of 108 dB when referred to 1 g full scale DC acceleration. The average noise floor in low-frequency range was less than −140 dBV. The ΣΔ micro-accelerometer system could achieve a resolution of 0.48 μg/Hz^1/2^ over a signal bandwidth. The test of the linearity is as shown in Figure 10b by the fitting of a straight line at ±1 g full scale. The ΣΔ micro-accelerometers could achieve a nonlinearity of 0.15% FS (full scale). After further electromagnetic shielding and vibration reduction, the output of the micro-accelerometer system was sampled when the sensor was at the state of zero acceleration in the laboratory test environment. The sampling time was longer than 4 h. After processing the sampled data with the Allen variance program in MATLAB, the bias stability test results of the closed-loop micro-accelerometer are shown as Figure 10c. The internal embedding plot in Figure 10c is processed sample data, and the bias stability is about 18 μg by calculation. We replaced 30 ASIC chips for the same sensitive structure and repeated the test. The bias stability of the closed-loop ΣΔ micro-accelerometer system was within 30 μg. Therefore, the micro-accelerometer system integrated with an ASIC chip had good output stability.

## 4. Conclusions

In this work, we proposed a high-order ΣΔ high-Q micro-accelerometer. In the ΣΔ interface ASIC, we used the correlated double sampling technique to eliminate the 1/*f* noise and offset for low-noise front-end detection. Additionally, the gain of the integrators was reduced by using the proportional scaling technique. The stability of high-order ΣΔ was studied by the root locus method. The interface circuit was fabricated in a standard 0.35 μm CMOS process. The test results of the system showed that: The micro-accelerometer could achieve a signal-to-noise ratio (SNR) of 108 dB; an average noise floor in low-frequency range of less than −140 dBV and a third harmonic distortion of −98 dB; a resolution of 0.48 μg/Hz^1/2^ (@300 Hz); a bias stability of 18 μg by the Allen variance program in MATLAB.

As shown in Table 2, the ΣΔ micro-accelerometer system could achieve a better performance than most of the reported accelerometers in Table 4.

We compared our work with the previously reported accelerometers based on a representative figure of merit (FOM = *P* × *a_n_* × *BW*^1/2^/*BW*), where *P* is the power dissipation, *a_n_* is the noise floor and *BW* is the signal bandwidth. This work is advantageous in the noise floor compared with [16,18,19] and a better FOM as shown in Table 2. We propose this interface ASIC based on the ΣΔ micro-accelerometer, which can satisfy the high-precision application in digital micro-accelerometers. The technical index of comprehensive performance can achieve a certain level.

## Figures and Tables

**Figure 1 micromachines-09-00675-f001:**
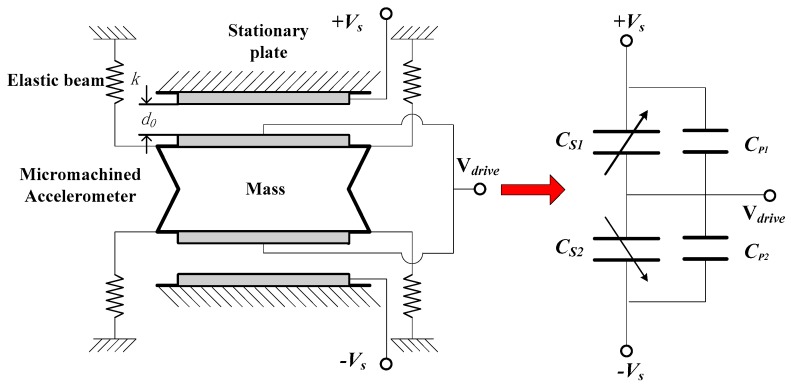
Vacuum-packaged bulk micro-accelerometer and equivalent bridge model.

**Figure 2 micromachines-09-00675-f002:**
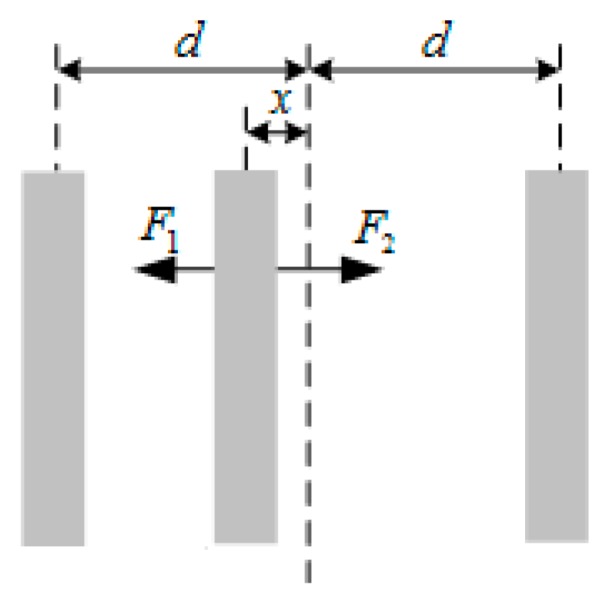
Differential capacitance model of the sensitive element.

**Figure 3 micromachines-09-00675-f003:**
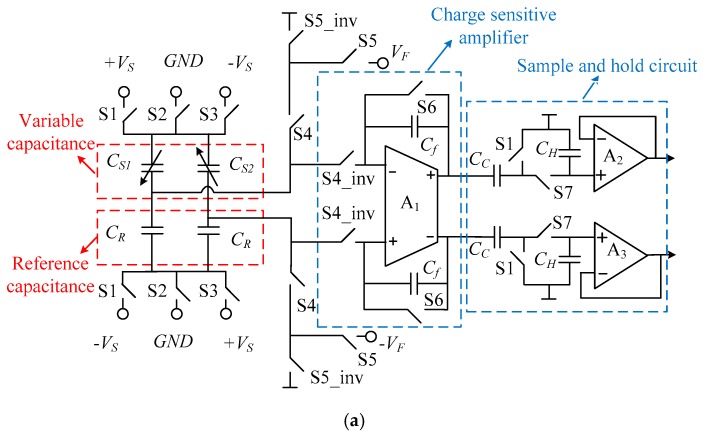
Front-end charge sensing circuit and timing diagram. (**a**) Front-end charge sensing circuit for micro-accelerometers; (**b**) Timing diagram for front-end charge sensing circuit.

**Figure 4 micromachines-09-00675-f004:**
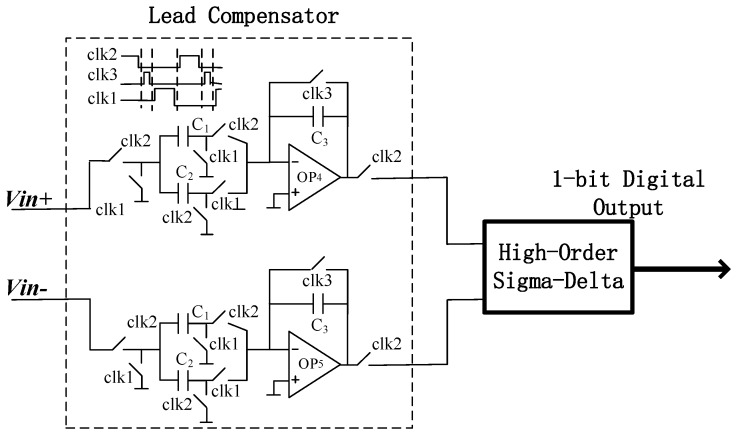
Lead compensator circuit.

**Figure 5 micromachines-09-00675-f005:**
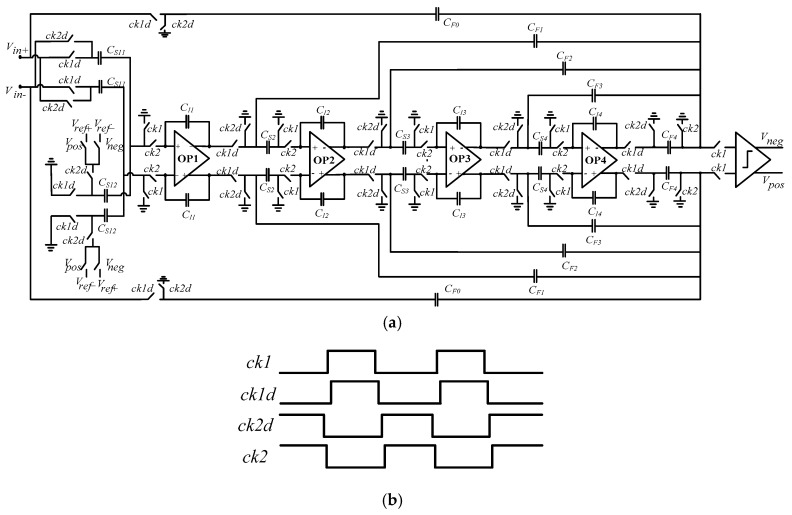
(**a**) High-order ΣΔ modulator circuit; (**b**) the timing diagram of ΣΔ modulator circuit.

**Figure 6 micromachines-09-00675-f006:**
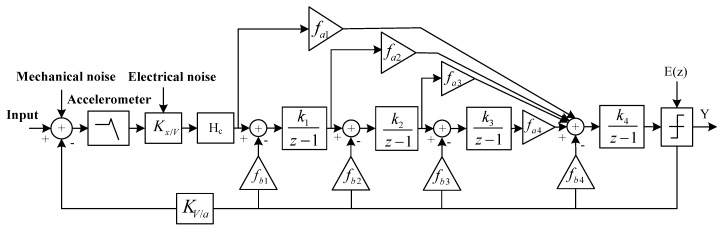
Analytical model of ΣΔ micro-accelerometers.

**Figure 7 micromachines-09-00675-f007:**
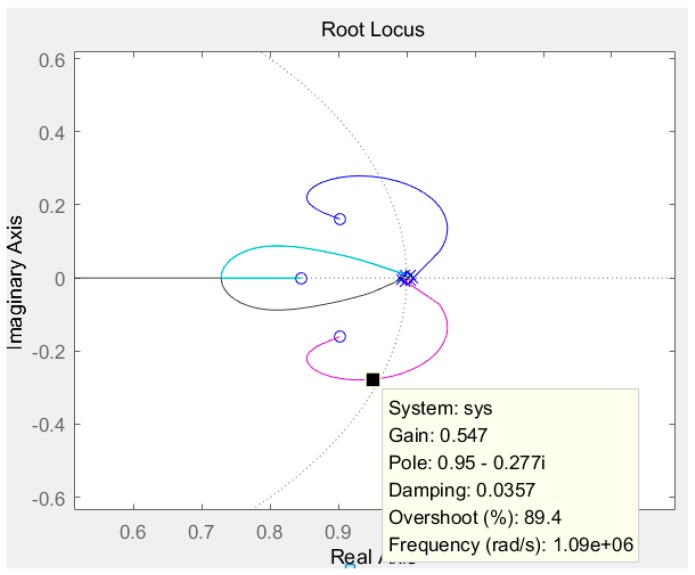
Root locus of the Sigma-Delta modulator.

**Figure 8 micromachines-09-00675-f008:**
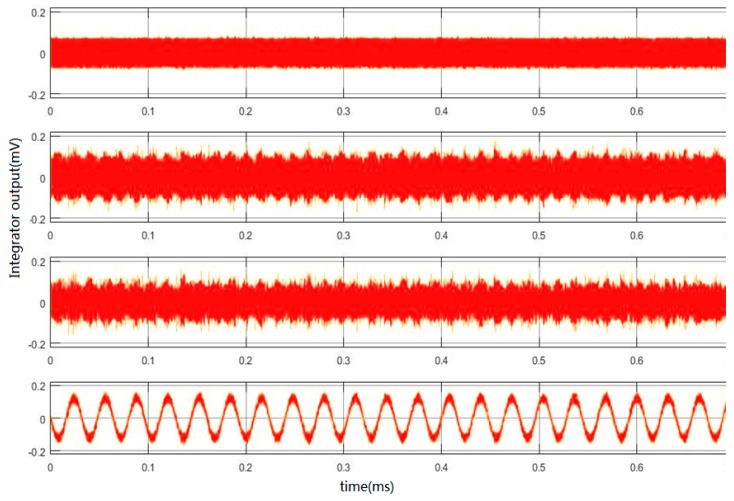
Output waves of each stage of the integrator.

**Figure 9 micromachines-09-00675-f009:**
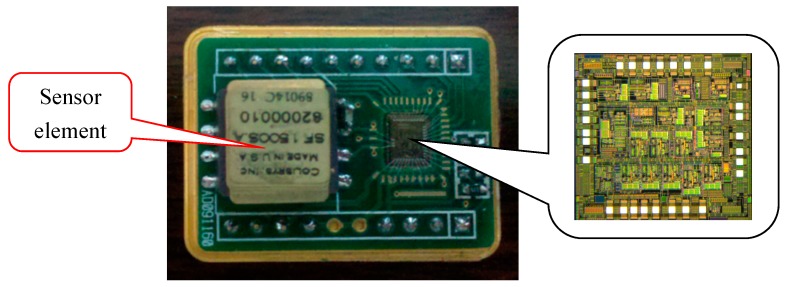
The printed circuit board photograph of ΣΔ modulator interface chip circuit

**Figure 10 micromachines-09-00675-f010:**
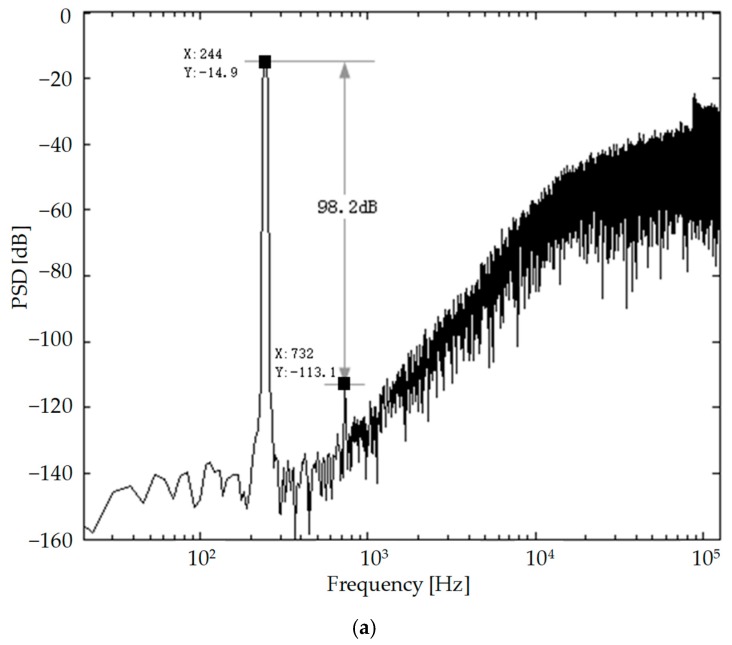
(**a**) The power spectrum density test of the digital accelerometer system; (**b**) the test of nonlinearity; (**c**) the test of bias stability.

**Table 1 micromachines-09-00675-t001:** Parameters of the high-*Q* sensor.

Parameters	Value
Sensitivity	10 pF/g
Proof Mass (*m*)	6.2 × 10^−7^ kg
Rest Capacitance (*C*_0_)	180 pF
Damping Coefficient (*b*)	0.01 N/(m·s)
Sensing Gap Distance (*d*)	2 μm
Resonance Frequency (ω_0_)	1000 Hz
Quality Factor (*Q*)	>30
Brownian Noise Floor	<60 ng/Hz^1/2^

**Table 2 micromachines-09-00675-t002:** Parameters of the ΣΔ modulator circuit.

ΣΔ Modulator Circuit	
Loop Filter Topology	Fourth-Order Switched-Capacitor
Integration Capacitor	10 pF
Oversampling Ratio (OSR)	417
Signal-to-Noise Ratio (SNR)	108 dB
Sampling Frequency	250 kHz
Third Harmonic Distortion	−98 dB

**Table 3 micromachines-09-00675-t003:** The modulator coefficient.

Coefficient	*k* _1_	*k* _2_	*k* _3_	*k* _4_	*f_a_* _1_	*f_a_* _2_	*f_a_* _3_	*f_a_* _4_	*f_b_* _1_	*f_b_* _2_	*f_b_* _3_	*f_b_* _4_
Value	0.05	0.8	0.2	0.05	0.4	0.2	0.1	0.4	0.2	0.3	0.5	0.6

**Table 4 micromachines-09-00675-t004:** Comparison of this work with other micro-accelerometers.

Parameters	[16]	[17]	[18]	[19]	This Work
Bandwidth (Hz)	200	300	500	300	300
Sensitivity (V/g)	0.495	2.267	NA	0.373	1.866
Noise floor (μg/Hz^1/2^)	2	0.3	4	1.15	0.48
Power (mW)	3.6	85.8	4.5	12	10
Process (μm)	0.35	0.7	0.5	0.6	0.35
Supply/Range	3.6 V/±1.15 g	5 V/±1.5 g	3 V/NA	9 V/±11 g	5 V/±1 g
Figure of Merit (FOM)	0.51	1.49	0.80	0.80	0.28

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
