# Peer review of "A High-Performance Digital Interface Circuit for a High-Q Micro-Electromechanical System Accelerometer"

_micromachines, 2018, doi:10.3390/mi9120675_

Round 1
Reviewer 1 Report
High-Q systems come with both benefits and drawbacks. It will be beneficial for readers to see an explanation in the introduction describing what will be compromised in high-q approach and if there is a pathway in this system to compensate for. (e.g. temperature sensitivity and circuit dynamic response)
It will be useful to add a plot of the PSD for quantization noise to Figure 9.
It is not clear what is the sensitivity after removing the loop gain?
Since the claims are made on the noise performance of the system, it will be very useful and fundamental to have Allan variance plot and analysis to show the accelerometer parameters.
In Table 2, it would be fair performance comparison only if the accelerometer sensitivity added to the figure-of-merit.
Author Response
Dear Reviewer:
Thank you for your letter and your comments concerning our manuscript entitled “A Novel High-precision Digital Tunneling Magnetic Resistance-type Sensor for the Nanosatellites’ Space Application”. Those comments are all valuable and very helpful for revising and improving my paper, as well as the important guiding significance to my researches. I have studied comments carefully and have made correction. We hope this time the paper can meet with approval. Revised portion are marked by green background in the paper.
Please find details in attachment.

Reviewer 2 Report
The paper describes a closed-loop digital interface circuit (front-end and delta-sigma modulator) for a high-Q MEMS accelerometer. The authors should note the following points:
Overall, the paper suffers from a lack of sufficient detail about the design procedure and experimental results. The authors are advised to add detailed analysis of the design in order to improve clarity for readers.
Fig. 1: Please add a labeled photograph of the accelerometer. A table summarizing its performance (sensitivity, proof mass, resonance frequency, etc.) would be useful as well.
Paragraph after eqn. (4): Please clarify in the text that you are denoting the input acceleration signal and the mechanical resonance frequency by "a" and "omega_0", respectively.
For clarity, please add a timing diagram of the switching waveforms (S1-S7) to Fig. 3.
Section 2: Please specify the design parameters of the delta-sigma modulator: oversampling ratio (OSR), signal-to-quantization noise ratio, loop filter topology, integration capacitor values, etc.
Section 3.1: Typo: "the Brown noise" -> "the Brownian noise".
Please clarify what parameter in Fig. 5 is changed in order to draw the root locus shown in Fig. How does this parameter vary with the input signal amplitude? Also, please clarify the conclusions that should be drawn from the root locus plot.
Fig. 7: Please specify the simulation conditions in the caption (supply voltage, input amplitude and frequency, etc.). Also, please use another color to plot the waveforms - yellow is very hard to see.
Fig. 8: Is the accelerometer shown on this figure? If so, please label it. If not, then please add it.
Sections 3.2 and 4: "third harmonic distortion of -98 dB" -> Please clarify that the distortion is calculated with respect to the fundamental component.
Sections 3.2 and 4: "average noise floor .. is less than -140 dB" -> What are the units being used here?
Sections 3.2 and 4: "resolution of ... over a signal bandwidth" -> What is the signal bandwidth?
Fig. 9(a): Please clarify the test conditions used to generate this plot. What was the input amplitude? Was this amplitude chosen in order to obtain the maximum SNR?
Fig. 9: Axes and figure labels are too small, please make them larger. Also, why does Fig. 9(c) have a box around it?
Author Response

(The authors gave the same response as above.)

Round 2
Reviewer 1 Report
Thanks for the considering suggestions and corrections.
Here are a few minor suggestions:
1- Please use vector image format so the quality preserves when converting to PDF. (Fig 7, Fig 8 and Fig 9 needs to be high quality for final submission)
2- Please color code (use color for signals) the Fig.3(b) for clarity.
3- Please consistently use the same font for figure legends, titles and axis values. (ex. Fig 10 a,b,& c should be plotted using the same style)
Reviewer 2 Report
The authors have extensively revised the paper based on the first round of reviews. It is now ready for publication.